# Fibromyalgia Animal Models Using Intermittent Cold and Psychological Stress

**DOI:** 10.3390/biomedicines12010056

**Published:** 2023-12-25

**Authors:** Hiroshi Ueda, Hiroyuki Neyama

**Affiliations:** 1Department of Pharmacology and Therapeutic Innovation, Nagasaki University Graduate School of Biomedical Sciences, Nagasaki 852-8521, Japan; neyama.hiroyuki.5y@kyoto-u.ac.jp; 2Department and Graduate Institute of Pharmacology, National Defense Medical Center, Taipei 114201, Taiwan; 3Multiomics Platform, Center for Cancer Immunotherapy and Immunobiology, Graduate School of Medicine, Kyoto University, Kyoto 606-8507, Japan

**Keywords:** stress-induced chronic pain, lack of morphine analgesia, empathic pain, pain memory, radical treatment, sex difference

## Abstract

Fibromyalgia (FM) is a chronic pain condition characterized by widespread musculoskeletal pain and other frequent symptoms such as fatigue, sleep disturbance, cognitive impairment, and mood disorder. Based on the view that intermittent stress would be the most probable etiology for FM, intermittent cold- and intermittent psychological stress-induced generalized pain (ICGP and IPGP) models in mice have been developed and validated as FM-like pain models in terms of the patho-physiological and pharmacotherapeutic features that are shared with clinical versions. Both models show long-lasting and generalized pain and female-predominant sex differences after gonadectomy. Like many other neuropathic pain models, ICGP and IPGP were abolished in lysophosphatidic acid receptor 1 (LPAR_1_) knock-out mice or by LPAR_1_ antagonist treatments, although deciding the clinical importance of this mechanism depends on waiting for the development of a clinically available LPAR_1_ antagonist. On the other hand, the nonsteroidal anti-inflammatory drug diclofenac with morphine did not suppress hyperalgesia in these models, and this is consistent with the clinical findings. Pharmacological studies suggest that the lack of morphine analgesia is associated with opioid tolerance upon the stress-induced release of endorphins and subsequent counterbalance through anti-opioid NMDA receptor mechanisms. Regarding pharmacotherapy, hyperalgesia in both models was suppressed by pregabalin and duloxetine, which have been approved for FM treatment in clinic. Notably, repeated treatments with mirtazapine, an α2 adrenergic receptor antagonist-type antidepressant, and donepezil, a drug for treating Alzheimer’s disease, showed potent therapeutic actions in these models. However, the pharmacotherapeutic treatment should be carried out 3 months after stress, which is stated in the FM guideline, and many preclinical studies, such as those analyzing molecular and cellular mechanisms, as well as additional evidence using different animal models, are required. Thus, the ICGP and IPGP models have the potential to help discover and characterize new therapeutic medicines that might be used for the radical treatment of FM, although there are several limitations to be overcome.

## 1. Introduction

Fibromyalgia (FM) is characterized by chronic widespread musculoskeletal pain and comorbid symptoms such as fatigue, sleep disturbance, cognitive impairment, and mood disorder [1,2,3,4,5]. In the first definition of FM by the American College of Rheumatology (ACR) in 1990, patients must state ‘painful’ and not ‘tender’ to ≥11 of 18 tender points upon an approximate force of 4 kg [6,7]. This was later revised in 2010 and 2011 to replace the tender point examination with a systemic symptom-based assessment of widespread pain, fatigue, sleep, cognitive and somatic disturbance [8,9,10]. The ACR further evolved the criterion in 2016 and now defines FM as [11] (1) generalized pain, defined as pain in ≥4 of five regions; (2) pain that lasts for ≥3 months; (3) a widespread pain index (WPI) of ≥7 and a symptom severity scale (SSS) score of ≥5 (or a WPI of 4–6 and an SSS score of ≥9); and (4) a diagnosis of FM that is irrespective of other diagnoses and does not exclude the presence of other illnesses. The International Association for the Study of Pain (IASP) also proposed a new category of chronic pain, ‘nociplastic pain’, which is distinct from nociceptive and neuropathic pain [2,12,13], and FM is considered a representative of this. Although the patho-physiology of FM and nociplastic pain remains to be elucidated, it is described that the pain signaling initiated in the peripheral and central nervous systems is dysregulated [14,15,16].

In the US, there are only three FDA-approved pharmacological treatments for FM [17]. The three FDA drugs are pregabalin, duloxetine, and milnacipran [18,19,20,21,22]. Pregabalin helps fibromyalgia by reducing pain and improving sleep and fatigue. Many investigators report that this medication helps to improve their overall vitality. But pregabalin can have some side effects, including drowsiness, dizziness, peripheral edema, and dry mouth [23,24]. Duloxetine and milnacipran, which are known to inhibit norepinephrine and serotonin re-uptake mechanisms, also help to improve pain and promote an overall feeling of improvement in patients with fibromyalgia [4,25,26,27]. As these neurotransmitters are linked to emotion and mood, these medicines can have some side effects, including nausea, dry mouth, constipation, decreased appetite, drowsiness, increased sweating, and agitation [22,28,29,30]. Therefore, long-term treatments using these medicines have serious limitations. The ideal treatments would be mechanism-based and radical treatments to erase pain memory. For this purpose, we need the development of various animal models that share similar pathological and pharmacotherapeutic features with clinical evidence, which include the fact that FM patients are less responsive to nonsteroidal anti-inflammatory drugs (NSAIDs) [31] and opioids [32,33,34]. In addition, some studies infer peripheral small fiber neuropathy as well as central mechanisms as causative factors in FM [35,36,37,38].

Aiming for better diagnoses or therapies, several groups of investigators have developed animal models mimicking the clinical features in the patho-physiology and pharmacotherapy of FM. They are vagotomized animals [39], repeated muscular acid saline-induced generalized pain (AcGP) models [40], reserpine treatment models [41], and intermittent sound stress models [42]. In addition to these models, the authors developed two unique chronic pain models using intermittent cold stress (ICS)- or intermittent psychological stress (IPS)-induced generalized pain (ICGP or IPGP) models in mice by modifying/adjusting previously reported stress models in mice and rats, which produced analgesia [43,44,45,46] or hyperalgesia [45,47,48,49,50]. As the ICGP and IPGP models were found to share similar patho-physiological and pharmacotherapeutic features with clinical evidence, the authors proposed that these animal models could be useful in developing new medicines for FM. In this review, the authors describe detailed protocols for the ICGP and IPGP models, summarize the patho-physiological and pharmacotherapeutic findings, which closely mimic the clinical evidence in FM patients, and introduce several pharmaceutical examples that show radical treatments for erasing pain memory status in FM-like models.

## 2. Intermittent Cold Stress (ICS)-Induced Generalized Pain (ICGP) Model

### 2.1. Protocol [51]

In this protocol [51], male or female C57BL/6J mice weighing 18–22 g were used. These mice were kept in the room under the condition of 22 ± 2 °C and a humidity 60 ± 5% and fed ad libitum with a standard laboratory diet and tap water before use. The intermittent cold stress (ICS) was started with two mice per group being kept in a cold room at 4 ± 2 °C at 16:30 (first day or day 0). The trials using different numbers of mice in the same cage, ranging from one to eight mice, revealed that the number of mice in the same cage should be two. Mice were given food pellets and gelatin on the floor ad libitum. A medical refrigerator with a transparent slide-type door is preferable for the cold room in order to keep a similar light/dark conditions even in the cold room (Figure 1). The temperature inside the refrigerator should not exceed 6 °C to avoid unstable hyperalgesia/allodynia. The mice were placed on a stainless-steel floor to give them a quick temperature change, and they were covered by a plexiglass cage in the cold room. At 10:00 on the next morning (day 1), the mice were taken out and placed at the normal temperature of 24 ± 2 °C, leaving the stainless floor in the cold room. After 30 min, the mice were put in the cold room again for another 30 min. These alternate temperature change processes were repeated until 16:30, followed by the placement of mice in the cold room overnight. On the next day (day 2), the same treatments were performed. Finally, on day 3 (post-stress day 1/P1), the mice were taken out of the cold room at 10:00 and kept at room temperature for adaptation (at least 1 h later) before the nociception tests. As the spontaneous activities of mice in the cold room lowered, food pellets and gelatin as the water supply were placed on the floor of the case to maintain healthy conditions. Nociception tests were performed after an adaptation period of at least 1 h for the stable nociception score. The constant cold stress (CCS) model experiments were performed as a reference. In this paradigm, the mice were kept in the cold room without alternate temperature changes for three consecutive nights. After ICS or CCS, the body weight of the mice decreased by approximately 10% at P2 to P3. The body weight of treated mice reached the level of the unstressed control mice at P5. However, a small number of ICS- or CCS-treated mice (~2% of the total mice) sometimes showed a decrease in body weight of 20%. In this case, they were euthanized. All experiments were performed in compliance with the relevant laws and institutional guidelines. All procedures were approved by the Nagasaki University Animal Care Committee [51].

### 2.2. Assessment of Stress-Related Behaviors

The mice treated with ICS did not show any gross behavioral changes as early as 1 h after the transfer from 4 °C to 24 °C, as reported in [52]. Furthermore, the ICS group did not show any significant change in the spontaneous locomotor activity, which was measured in the open field (22 cm × 33 cm) for 3 min using a SCANET apparatus (Melquest, Toyama, Japan). In the study, when estimating anxiety using the elevated plus maze test, there was no significant change in the time spent in the open arm. Additionally, ICS did not cause any change in the total duration of immobility in the tail suspension test, in which mice were suspended 30 cm above the floor, suggesting that ICS had no depressive action. Thus, it is evident that the ICS group in this paradigm may have no behavioral changes, including depression, which is reported to co-exist with hyperalgesia [53]. However, when many more days of alternate temperature changes (ICS) are used, some depression-like behavioral changes may occur (unpublished data). In order to avoid such depression-like behaviors, the quick changes in temperature using a metal plate in the refrigerator are highly recommended.

### 2.3. Measurement of Plasma Corticosterone

Blood from the mice treated with ICS was collected at 21:00 to exclude the effect of circadian rhythms on circulating plasma corticosterone. Plasma corticosterone levels were estimated fluorometrically, according to the method of Zenker and Bernstein [54]. The plasma corticosterone levels slightly increased at P1, and after the ICS, they increased at P2. Significant changes in the plasma corticosterone levels were observed at P1, but no significant change was observed between P3 and P12 [52]. It is notable, however, that CCS showed no effect on the corticosterone levels between P1 and P12.

### 2.4. Long-Term and Generalized Pain

Long-lasting hyperalgesia was observed in three different nociception tests. In the mechanical paw pressure test, a mechanical stimulus was delivered to the middle of the plantar surface of the right or left hind paw using a digital von Frey Anesthesiometer^®^ with a Rigid Tip (Model 2390, 90 g probe, 0.8 mm in outer diameter, IITC Inc., Woodland Hills, CA, USA) and a Transducer Indicator (Model 1601; IITC Inc., Woodland Hills, CA, USA). The mice were placed on a 6 mm × 6 mm wire mesh grid floor and allowed to acclimate for a period of 1 h. The pain threshold was defined by measuring the pressure threshold to induce a withdrawal flexor response. A cut-off pressure of 20 g was set to avoid tissue damage [51,52,55]. Significant mechanical hyperalgesia (or allodynia) on either the right or left hind paw was found at P1 after the ICS, and a similar level of hyperalgesia lasted until at least P13 [51,55]. On the other hand, following CCS, significant hyperalgesia was also observed at P1, but no hyperalgesia was observed at P5 or later [51].

In the mechanical muscle pain test, a Randall–Selitto-type pressure analgesia meter MK-201D (Muromachi Kikai, Tokyo, Japan) was used [56]. In this method, the probe was applied to the right femoral muscle, and the threshold to show struggling behavior was measured. The threshold (on average) was decreased from ~120 mmHg to ~70 mmHg at P5 after the ICS. Thus, as the ICS-specific hyperalgesia at P5 or later was observed not only in both sides of the paw but also in the femoral muscle, the hyperalgesia may be called ICS-induced generalized pain (ICGP).

On the other hand, in Hargreaves’s plantar test, the nociception threshold was assessed using the latency of paw withdrawal upon a thermal stimulus. In this thermal test, unanesthetized mice were placed in a plexiglass cage on top of a glass sheet and acclimated for 1 h. The projection bulb of the thermal stimulator (IITC Inc., Woodland Hills, CA, USA) was positioned under the glass sheet, and the focus was aimed exactly at the middle of the plantar surface of a mouse by using a mirror attached to the stimulator. A cut-off time of 20 s was set to prevent tissue damage. At the beginning of a series of experiments, the distance between the plantar surface and projection bulb and the intensity of the beam were adjusted to make the control threshold around 10 s. In this apparatus, the regulator was set at an intensity of 20, which increases the plantar surface temperature to 45.1 ± 1.3 °C (*n* = 10) 10 s after the start of thermal stimulation [57]. Thermal hyperalgesia following the ICS lasted until at least P17 or later [55].

In the electrical-stimulation-induced paw withdrawal (EPW) test using a Neurometer^®^ Current Perception Threshold/C (CPT/C; Neurotron Inc., Baltimore, MD, USA) [58,59], electrodes were attached to the planter and the insteps of the hind paw, and transcutaneous nerve stimuli with sine-wave pulses of 5, 250, or 2000 Hz were applied. The current threshold was defined by the minimum intensity (μA) at which each mouse showed paw withdrawal behavior. The in vivo patch-clamp recording studies characterized the sensory fiber specificity for C, Aδ, or Aβ fibers. In this electro-physiological study using the acutely isolated rat spinal cord slice, excitatory synaptic responses in the substantia gelatinosa neurons evoked by transcutaneous stimuli at 5, 250, or 2000 Hz, respectively, were evaluated [60]. The minimum intensity (μA) at which a mouse withdrew a paw was defined as the current threshold. The threshold to show paw withdrawal behavior by C fiber (5 Hz) stimulation was around 100 μA, but no significant change was observed for ICS. On the other hand, ICS significantly decreased the threshold for Aδ (250 Hz) or Aβ (2000 Hz) from 240 to 130 μA or 360 to 220 μA, respectively [57]. As the nociceptive withdrawal responses were caused by subthreshold stimulation, these abnormal pain behaviors could be considered as allodynia. Furthermore, the allodynia in the ICGP model was specific to the stimulation of Aδ or Aβ fibers, but not C fiber.

### 2.5. Sex Difference

There was no significant difference in the ICS-induced mechanical hyperalgesia between male and female mice [51]. However, significant female-predominant ICS-induced hyperalgesia was found in mice with gonadectomy, which was performed as reported in [61]. After the surgery, the mice were kept in a soft bed cage with some food inside. The ICS treatment was given 3 weeks after the orchiectomy (ORX) or ovariectomy (OVX). The pretreatment with gonadectomy had no significant influence on either the gross behavioral activities or the control nociceptive threshold in the paw pressure test in male or female mice. However, the ORX pretreatment significantly decreased the ICS-induced mechanical hyperalgesia in male mice, while the OVX pretreatment did not affect the ICS-induced hyperalgesia.

### 2.6. Involvement of Lysophosphatidic Acid (LPA) Receptor Signaling

Since the first discovery that lysophosphatidic acid receptor (LPAR) signaling initiates nerve injury-induced neuropathic pain (NeuP) [62], many studies revealed that LPAR_1_ plays key roles in the development and maintenance of various NeuP models, including partial sciatic nerve ligation-, paclitaxel-, spinal cord injury-induced, type 1 diabetic, type 2 diabetic, and central, post-stroke neuropathic pain models [63,64,65]. Similarly, in LPAR_1_-knock-out (KO) mice, ICS-induced thermal hyperalgesia was completely lost, with no difference in the basal pain threshold compared with wild-type mice [56].

### 2.7. Pharmacotherapy

The FDA has approved three medicines to treat FM. They are antidepressants—duloxetine (Cymbalta) and milnacipran (Savella)—and the anti-seizure medicine pregabalin (Lyrica). Although duloxetine and pregabalin have worldwide approval for the treatment of FM, milnacipran is only licensed in Canada, Russia, and the United States of America. However, drugs such as nonsteroidal anti-inflammatory drugs (NSAIDs), opioids, and corticosteroids have not been found to be effective in FM patients [29,32].

#### 2.7.1. Morphine

Systemic (s.c.) injection of morphine hydrochloride (0.3 to 3 mg/kg) dose-dependently increased the analgesic potency in the thermal nociception test in naive mice, and significant actions were observed at 1 and 3 mg/kg (s.c.) at P5 [66]. In ICS-treated mice, however, no significant morphine analgesia was observed at 3 mg/kg (s.c.). Similarly, when morphine was administered intra-cerebroventricularly (i.c.v.) at 0.03 to 0.3 nmol, there was dose-dependent and significant analgesia in the naive mice, but no significant analgesia was observed at 0.3 nmol (i.c.v.) at P5 in the ICS model. It should be noted that there was significant analgesia at 0.3 nmol (i.c.v.) at P5 in the CCS model. However, the intra-thecal (i.t.) injection of morphine showed significant analgesia in naive and ICS-treated mice. These findings suggest that ICS may lose morphine-analgesia mechanisms in the brain, which drives a descending pain-inhibitory system to the spinal dorsal horn. This view was supported by the finding that the ICS treatment abolished the morphine (30 nmol, i.c.v.)-induced increase in the turnover rate of serotonergic neurons [5-HIAA]/[5-HT] in the spinal dorsal horn [66].

#### 2.7.2. Antidepressants

As gabapentin and pregabalin, which are widely used to treat FM patients in the clinic, alleviate abnormal pain and the accompanying fatigue and insomnia without affecting depressive symptoms [67,68], the presence of depression-like behavior is unlikely to be necessary in animal models of FM. In that respect, the ICS model seems to be preferable for the evaluation of FM-like hyperalgesia since this model did not show any significant changes in the tail-suspension test, which is used for the evaluation of depression-like behavior [49]. Recently, milnacipran and duloxetine, which are serotonin/norepinephrine re-uptake inhibitors, have been approved by the United States Food and Drug Administration (FDA) for treating FM pain. The anti-hyperalgesia actions of these compounds in clinic for FM patients are presumably attributed to the enhancement of central descending monoaminergic pathways in pain inhibition [69,70]. Indeed, the intra-thecal injection of anti-depressants, which have potent serotonin and/or norepinephrine re-uptake inhibitors (SNRIs), showed potent anti-hyperalgesia actions in the ICS model [52]. Equivalent anti-hyperalgesia actions were observed with milnacipran, amitriptyline, mianserin, and paroxetine at 0.1, 15, 15, and 10 μg (i.t.), respectively. These results indicate that milnacipran possesses the re-uptake inhibitory action of norepinephrine, as well as serotonin, and was most potent. It should be noted that repeated i.t. treatments with these anti-depressants produced complete relief of ICS-induced hyperalgesia [52].

Mirtazapine is an anti-depressant, and the pharmacological mechanisms underlying anti-depression activity in it are reported to be associated with the disinhibition of serotonin release by blocking the pre-synaptic adrenergic α_2_ receptor on serotonin and norepinephrine neurons, as well as blockading postsynaptic excitatory 5-HT_2_ and 5-HT_3_ receptors so that the 5-HT_1_ receptor-mediated inhibitory signal becomes predominant [71,72]. When mirtazapine at 1 mg/kg (i.p.) was given, a significant reversal of thermal hyperalgesia was observed [55]. The peak effect was observed at 30 min and disappeared at 3 h. On the other hand, the i.c.v. injection of mirtazapine at 1 μg showed a longer anti-hyperalgesia effect. The peak effect was observed at 1 h, and 80% of maximal anti-hyperalgesia effects remained at 3 h. However, no significant anti-hyperalgesia effect was observed with the intra-thecal treatment at 1 μg. Similar results were also observed in the mechanical paw pressure test [55]. When mirtazapine (1 mg/kg, i.p.) was given in ICS-treated mice every other day from P5 to P13, the basal nociceptive threshold gradually reversed to the control level. The complete reversal of hyperalgesia lasted at least until P17, suggesting that so-called “pain memory” was erased with repeated mirtazapine treatments.

Notably, the repeated treatments with mirtazapine by i.p. or i.c.v. administration recovered the morphine (i.c.v.) analgesia in the ICS model [55]. In order to investigate the site of action for mirtazapine-induced anti-hyperalgesia actions and the recovery of morphine analgesia, the ICS-induced changes in the gene expression of target receptors, such as adrenergic α_2_, serotonergic 5HT_2_, 5-HT_3_, and histaminergic H_1_ receptors, were performed [72]. Among 16 pain-related brain regions, only habenula showed a time-dependent increase in the transcription of adrenergic α_2_a receptor (ADRA2A), which started at P1 and lasted through P12 after ICS exposure [55]. When the siRNA for ADRA2A was bilaterally micro-injected (0.5 μL each) into the habenula at P3 after the ICS, a significant reversal of hyperalgesia was observed at P5, whereas the loss of morphine (i.c.v.) analgesia was not affected. However, morphine analgesia was recovered by the repeated i.c.v. injection of ADRA2A siRNA. All these findings suggest that the loss of morphine analgesia in the ICS model is independent of the mechanisms underlying hyperalgesia, but the mirtazapine-induced recovery of morphine analgesia seems to be related to unidentified actions in brain regions different from the habenula.

#### 2.7.3. Pregabalin (PGB)

An initial pharmacotherapeutic study into the ICS model was carried out with gabapentin [51], which has been widely used as an off-label drug for chronic pain treatment. In this study, the systemic administration of gabapentin dose-dependently reversed the ICS-induced mechanical hyperalgesia in the range of 0.3–3 mg/kg (i.p.). Significant actions were observed at 1 or 3 mg/kg, which is much lower than the case (30 mg/kg) in the partial nerve ligation (pSNL)-induced NeuP. Of most interest is the finding that the i.c.v. treatment with this drug at 3 μg showed significant anti-hyperalgesia action in the ICS model, while pSNL-hyperalgesia was not suppressed by 30 μg (i.c.v.). This finding suggests that gabapentin may have specific molecular targets in the brain with ICS treatment.

PGB, a structural derivative of gamma aminobutyric acid (GABA), was approved by the FDA for the management of NeuP in 2004 and FM in 2007. Similar to the case with gabapentin, the anti-thermal hyperalgesia action of PGB in the ICS model was observed at 1 mg/kg (i.p.), which is much lower than the case (PGB, 30 mg/kg, i.p.) in the pSNL model [73]. Similarly, significant anti-hyperalgesia effects from PGB (1 μg, i.c.v.) were observed up to 72 h in the ICS model, while no significant effects from PGB (30 μg, i.c.v.) were observed in the pSNL model. It should be noted that PGB showed potent and long-lasting anti-hyperalgesia effects via the intra-thecal injection (30 μg, i.t.) up to 24 h [73], suggesting that the site of PGB in the pSNL model is in the spinal cord or DRG. In contrast, PGB (10 μg, i.t.) showed significant but less potent anti-hyperalgesia effects in the ICS model when compared with the i.c.v. injection case. Notably, when PGB was administered i.c.v. post-ICS at 5, 8, and 11 days (P5, 8, and 11), the nociceptive threshold was reversed on P14 to the control level without ICS, and it lasted until at least P20 [73]. This finding indicates that the established pain memory in the ICS model could be reversible by using repeated PGB (i.c.v.) treatments. As the central injection of PGB is not practical in clinic due to invasiveness, the author’s group performed the combinatorial treatment with valspodar, a non-immunosuppressive P-glycoprotein inhibitor, which reverses multidrug resistance [74,75,76]. When PGB (1 mg/kg, i.p.) and valspodar (3 mg/kg, i.p.) were co-administered at P5, 7, 9, and 11, the nociceptive threshold was reversed to the control level on P13, and it lasted at least until P19 [73]. However, there is no evidence that the accumulation of PGB in the brain is increased by the presence of valspodar.

#### 2.7.4. Donepezil

The new guidelines approved by the American College of Rheumatology [77] state that the criteria for FM include the severity of somatic symptoms such as fatigue, inability to experience refreshing sleep, waking up tired and developing cognitive disturbances, as well as widespread pain. Many studies have also shown that FM patients also experience dryness of cornea or eyes and mouth [78]. Based on such somatic symptoms, the authors attempted to use pilocarpine, a cholinergic agonist, in the ICS-type FM model [79]. As was the case with mirtazapine and PGB, pilocarpine showed potent anti-hyperalgesia effects at 1 mg/kg (i.p.) and 1 μg (i.c.v.) but not 1 μg (i.t.). As the anti-hyperalgesia action by pilocarpine (1 mg/kg, i.p.) was abolished by pirenzepine (1 μg, i.c.v.), a muscarinic receptor antagonist, it is evident that brain muscarinic receptor mechanisms play major roles in the pilocarpine (i.p.)-induced suppression of ICS-induced hyperalgesia.

Of most interest is the finding that similar anti-hyperalgesia effects were observed with a small dose of donepezil [79], which readily penetrates the brain and increases brain acetylcholine levels by inhibiting acetylcholine esterase [80]. Donepezil at 10 μg (i.p.) reversed the hyperalgesia to the control level at 1.5–2 h in the thermal and mechanical nociception tests. Similar beneficial effects were also observed with the i.c.v. injection (10 μg) but not with the i.t. injection (10 μg). As atropine at 100 ng (i.c.v.) also abolished the anti-hyperalgesia donepezil effects at 10 μg (i.p.), the beneficial effects of donepezil appear to be mediated through an activation of the brain muscarinic receptor system. As in the cases with mirtazapine, PGB, and pilocarpine, repeated donepezil treatments (daily) between P5 and 10 reversed levels to the normal threshold at P12.

## 3. Intermittent Psychological Stress (IPS)-Induced Generalized Pain (IPGP) Model

### 3.1. Protocol [56]

In this pain model, mice were exposed to intermittent psychological stress or empathic stress. Mice weighing 20–25 g were put in the communication box (CBX-9M, Muromachi-Kikai, Tokyo, Japan) that has nine compartments (10 cm × 10 cm), which were divided with transparent plexiglass walls (Figure 2). Five mice were put in the compartments that had the grid floor for electrical shocks, while four other mice were put in the remaining compartments, in which the grid floors were covered with insulating plastic plates. The foot shock was given to five mice by a shock generator (CSG-001, Muromachi-Kikai, Tokyo, Japan) and cycler timer (CBX-CT, Muromachi-Kikai, Tokyo, Japan) [81,82], and psychological stress from foot-shock compartments measured in terms of vision, hearing, and smell was given to four mice. For the repeated foot shock stress (RFS), a short duration of electrical shock (0.6 mA, 1 s) was delivered to the feet, halting every 47 s (120 times), whereas for the randomly programmed intermittent foot shock stress (IFS), the same number of shocks were delivered, taking 24–96 min. The other groups without foot shocking were called repeated psychological stress (RPS) and intermittent psychological stress (IPS). In all experimental paradigms, the stress was given once per day for 5 days. All the procedures used in this work were approved by the Nagasaki University Animal Care Committee and were complied with the fundamental guidelines for the proper conduct of animal experiments and related activities in academic research institutions under the jurisdiction of the Ministry of Education, Culture, Sports, Science and Technology, Japan [56].

### 3.2. Behavioral Observation

The mice given electrical foot shocks (RFS and IFS) showed nociceptive behaviors, such as rearing, jumping, and vocalization, whereas the other mice without foot shock (RPS and IPS) were supposed to keep watching and smelling the mice that were given foot shocks, listening to their vocalization over the plexiglass walls. The mice without foot shocks (RPS and IPS) looked restless without significant behavioral abnormalities throughout the foot shock or psychological stress paradigm. IPS showed neither depression-related behaviors during the immobility time in the tail-suspension test nor in the forced swimming test [56].

### 3.3. Corticosterone Levels

The plasma corticosterone levels in the mice with IFS and IPS were observed. The elevation in corticosterone levels at 20:00 started at P1, reached a maximum at P2, then slightly declined at P5, and returned to the control level by P12 in both the IFS and IPS-treated mice [56]. The peak level at P2 was approximately 3.5- and 2.2-fold that of the control level (10 μg/dL) in the IFS- and IPS-treated mice. When the plasma levels of the IFS- and IPS-treated mice were measured at 08:00 on P2, the values were 10 and 5 μg/dL, respectively.

### 3.4. Long-Term IPGP

In the mechanical paw pressure test using an electronic digital von Frey Anesthesiometer^®^ and Rigid Tip, the mice treated with IPS, IFS, RPS, and RPS all showed significant hyperalgesia/allodynia on day 1 (P1) after stress [56]. However, only the IPS-treated mice showed significant hyperalgesia at P8. The fact that simply repeating the psychological stress (RPS) did not show long-lasting hyperalgesia suggests that randomly programmed intermittent or unexpected stress causes more stable pain behaviors than simply repeated psychological stress. Similar results were also found in the thermal paw withdrawal test. The mice with IFS showed significant hyperalgesia when given stress on day 1 (D1), and it chronologically increased until D5 (post-stress day 0/P0), but the threshold was then gradually returned to the control level at P6 and later. In contrast, the mice with IPS showed significant hyperalgesia at D3, and more potent and significant hyperalgesia was observed from D5 to at least P19 without attenuation. As mentioned above, the plasma corticosterone levels were lower in the IPS-treated mice than in the IFS-treated mice. Therefore, long-term hyperalgesia is unlikely to be associated with the quantitative stress level, but the difference may be caused by unidentified qualitative differences in the stress effects.

In other nociceptive tests, mechanical muscle pain was enacted by using a Randall–Selitto-type pressure analgesia meter, MK-201D (Muromachi Kikai, Tokyo, Japan), which displays the pressure digitally. For the evaluation of muscle pain threshold, the pressure threshold (in mmHg) that causes struggling behavior in the right femur was measured. The IFS-treated mice showed significant hyperalgesia at P0 and P3 but not P7, whereas the IPS-treated mice showed consistent hyperalgesia from P0 to P7 without attenuation [56]. In the chemical nociception test, which was called the acetic acid-induced writhing test, the number of typical writhing behaviors after the intraperitoneal injection of 0.9% acetic acid solution (0.1 mL per 10 g of body weight) was counted for 20 min post-injection. In this test, the amount of writhing in the IPS-treated mice significantly increased [56]. All these results suggest that IPS induces chronic generalized pain (IPGP). However, the IPS-treated mice had no effect in the visceral pain test, in which 20 μL of 0.3% capsaicin solution dissolved in 10% ethanol and 10% Tween-80 in physiological saline was injected into the colon, and the licking of the abdomen was measured for 20 min [56]. These findings suggest that IPGP is observed upon somatic but not visceral pain stimulation.

### 3.5. Sex Difference

There is no significant difference in the mechanical nociceptive threshold between naive male and female mice. Both male and female mice showed a similar threshold of approximately 10 g. In male mice, the IPS treatment decreased the threshold to ~6 g and ~7 g at P1 and P15, respectively. In female mice, on the other hand, the same treatment decreased the threshold to ~5 g and ~5.5 g at P1 and P15, respectively [56], indicating that the IPS-induced hyperalgesia in female mice is slightly higher than that in male mice. However, a remarkably evident difference in terms of sex in IPS-induced hyperalgesia was observed after the pretreatment with gonadectomy by removing the testes (orchiectomy/ORX) or ovaries (ovariectomy/OVX) 3 weeks prior to IPS treatment [56]. The threshold in ORX-treated mice was chronologically recovered from P1 to P15. At P15, hyperalgesia was completely abolished. On the other hand, the threshold in OVX-treated mice was unchanged throughout P1 to P15. Although the mechanisms underlying the gonadectomy-specific sex difference remain elusive, there is a possibility that androgens may play roles in the maintenance of IPGP in male mice, while female mice in nature may have unidentified maintenance mechanisms for IPGP.

### 3.6. Involvement of LPAR_1_ Signaling

IPS-induced thermal hyperalgesia at P1 was lost in the LPAR_1_-KO mice [56]. The repeated i.c.v. treatments with 1 nmol of AM966, an LPAR_1_ antagonist, from P5 to P11 abolished the established hyperalgesia at P12, whereas AM966 (i.c.v.) showed no change in the thermal threshold throughout 3 h in the control mice or IPS-treated mice at P5 [56]. These results suggest that LPAR_1_ signaling plays key roles in the development and maintenance of IPGP.

### 3.7. Monoamine Turnover Rate

Descending norepinephrine (NE) and serotonin (5-HT) neuronal systems from the PAG (periaqueductal gray) or RVM (rostral ventromedial medulla) to the spinal cord play roles in morphine analgesia [83]. In addition, it is reported that the lack of morphine analgesia may be related to the findings that fibromyalgia patients have decreased levels of spinal cord monoamines [84]. In accordance with these reports, the turnover rate of NE neuronal activity, which is calculated as [MHPG levels]/[NE levels] in the spinal cord of IPS mice, significantly decreased compared to the rate in the control mice [56]. In this study, NE and its metabolite MHPG (3-methoxy-4-hydroxyphenylglycol) were separated by high-performance liquid chromatography and measured by using an electrochemical detector. Although there was a slight decrease, no significant change in the turnover rate of 5-HT or dopamine neuronal activities was observed.

### 3.8. Pharmacotherapy

#### 3.8.1. Lack of Actions of Diclofenac and Morphine

Diclofenac, a representative NSAID, at 10 mg/kg (i.p.) did not show any anti-hyperalgesia effects in the thermal nociception test at P5 in the IPS model, although the same dose of diclofenac showed potent anti-hyperalgesia actions in the inflammatory pain model using Freund’s complete adjuvant [56]. Similarly, the treatment with morphine at 0.3 nmol (i.c.v.) did not show any analgesia or anti-hyperalgesia in the IPS model [56]. The lack of morphine analgesia was also observed when it was given subcutaneously. These findings are consistent with the case of the ICS model [66].

#### 3.8.2. Pregabalin (PGB)

PGB at 0.3 to 3 mg/kg (i.p.) dose-dependently showed anti-hyperalgesia effects in the IPGP model but not in control mice [56]. The peak effect was observed at 0.5 h. On the other hand, the brain injection of PGB at 0.3 μg (i.c.v.) showed significant anti-hyperalgesia effects up to 48 h. The complete reversal of hyperalgesia was observed from 1 to 24 h after the treatment. However, the i.t. injection with PGB at 3 μg showed a significant but weak and short-term anti-hyperalgesia effect only at 0.5 h. The brain-specific PGB effects were also observed in the ICGP model [73].

#### 3.8.3. Duloxetine

Duloxetine (DLX), a NE and 5-HT re-uptake inhibitor (SNRI), at 30 mg/kg (i.p.) showed significant anti-hyperalgesia effects from 0.5 to 3 h in the IPGP model but not in the control mice [56]. The intra-thecal treatment with DLX at 1 μg showed significant anti-hyperalgesia at 0.5 h, whereas no significant anti-hyperalgesia was observed with DLX at 3 μg (i.c.v.), which is consistent with the findings in the ICGP model, in which several NE/5-HT re-uptake inhibitors (i.t.) showed potent hyperalgesia [52].

#### 3.8.4. Mirtazapine

Mirtazapine, a novel type of antidepressant [72,85], at 1 mg/kg i.p. showed significant anti-hyperalgesia effects at 0.5 and 1 h but not 3 h in the thermal nociception test in the IPGP model. When mirtazapine at 1 μg was given through an i.c.v. route, a complete reversal of hyperalgesia from 0.5 to 3 h and significant anti-hyperalgesia at 24 h was observed in the IPGP model but not in the control mice [56]. As seen in the case with the ICGP model [55], no significant anti-hyperalgesia was observed with mirtazapine at 1 μg (i.t.).

## 4. Detailed Mechanisms of ICGP and IPGP

### 4.1. ICGP and IPGP Model Mimic the Patho-Physiological Features of FM

In the ICGP model, we used the stainless steel floor inside of the refrigerator, which is very important to produce a quick temperature change and show long-term hyperalgesia by at least P17 or later [55] without depression-type behaviors. The use of a refrigerator is also preferred by the experimenter, who will not experience the stress of similar alternate temperature changes. The ICS is a modified stress model for the specific alteration of rhythm in environmental temperature (SART), which causes hyperalgesia. However, the hyperalgesia induced by SART stress, in which the experimenters transfer the mouse cage between 4 °C and 24 °C rooms, recovers 4 days after the last exposure to cold stress [50]. In some experiments, using SART stress for longer (over 5 days), hyperalgesia may continue longer, but, in this case, depression-related behaviors will emerge. From these points, the authors call this modified stress model the intermittent cold stress (ICS)-induced generalized pain (ICGP) model, being distinct from the SART model. In addition, we specify stable hyperalgesia in comparison to constant cold stress, which shows a disappearance of significant hyperalgesia by day 5 after the last dose of cold stress [50].

On the other hand, there are reports that analgesic or hyperalgesic responses are observed with vibration under environmentally different (quiet or agitated) conditions [86], or with foot shocks under DURA-intact or DURA-ligated conditions, respectively [45]. Based on such findings, we searched for a stress condition to produce stable hyperalgesia. The best result was obtained by using a nine-compartment-type community box, where four mice experience empathy in response to the surrounding five foot-shocked mice, which receive randomized electrical stimulation [56]. Both the ICGP and IPGP models share the patho-physiological and pharmacotherapeutic features observed in FM patients (Table 1). Regarding patho-physiology, both model mice show long-term hyperalgesia in the thermal and mechanical nociception tests. In the EPW test using electrical paw stimulation, the ICGP mice showed hyperalgesia in response to the stimuli at 250 (Aδ) and 2000 Hz (Aβ) to the paw and to the pressure stimuli at the femoral muscle [57]. Similarly, the IPGP mice also showed hyperalgesia responses to the pressure stimuli at the femoral muscle and to the chemical stress with acetic acid in the abdomen [56]. These features indicate that the hyperalgesia experienced was widespread.

However, there was no hyperalgesia in the visceral pain model, in which the intra-colon injection of capsaicin was used in the IPGP model. Although the lack of hyperalgesia in visceral pain remains elusive, it is presumed that the spinal-to-brain pain pathway following somatic and visceral pain stimuli is different, and IPS does not affect the visceral pathway. Both of the model mice showed mechanical and thermal hyperalgesia, in contrast to the AcGP model, which does not show thermal hyperalgesia. The mechanisms underlying the lack of thermal hyperalgesia in the AcGP model remain elusive, and this possibly suggests an alternate central pain pathway. Regarding the female-predominant sex difference, the studies with ICGP and IPGP models showed a remarkable sex difference after the gonadectomy [56,87]. It should be noted that there was only a slight difference in the hyperalgesia induced by ICS or IPS between both the male and female mice, but the hyperalgesia in male mice with gonadectomy was completely recovered 15 days later. Male mice may have some resilient function to keep a normal pain threshold after gonadectomy. In other words, androgens may maintain hyperalgesia status.

### 4.2. ICGP and IPGP Model Mimic the Pharmacotherapeutic Features of FM

In accordance with the clinical report of no participant-reported pain relief in FM patients with NSAIDs [29], diclofenac, a representative NSAID, did not show any anti-hyperalgesia in the IPGP model [56]. This fact is easy to understand since no tissue inflammation was detected in this model. In addition, the lack of analgesia as a result of morphine given through the systemic and i.c.v. routes in the ICGP and IPGP model is consistent with the report that there is no conclusive evidence that opioids are effective in the treatment of FM in clinic [32]. On the other hand, anti-convulsant PGB, the serotonin and norepinephrine re-uptake inhibitor duloxetine, and FDA-approved milnacipran are widely used for the treatment of FM in clinic. In the IPGP and/or ICGP model, all these compounds showed potent anti-hyperalgesia actions. It should be noted that when PGB was given systemically (i.p.), it showed more potency in the ICGP and IPGP models than in the pSNL-induced NeuP model [73]. A more striking finding was the fact that PGB (1 μg, i.c.v.) showed significant anti-hyperalgesia action for 3 days, whereas it did not show any significant effect on the pSNL model at 30 μg (i.c.v.) in the ICGP model [73], suggesting that PGB may have additional sites of action to α2δ in the brain of the ICGP or IPGP model. Milnacipran and duloxetine showed potent anti-hyperalgesia effects when given i.t. in the ICGP and IPGP model, respectively [52,56]. These actions are presumably attributed to the elevation in the descending pain-inhibitory serotonin and norepinephrine neuronal system. Indeed, the IPGP model showed a decrease in the norepinephrine turnover rate [56]. In the ICGP model, on the other hand, the central morphine-induced elevation of serotonin turnover rate in the spinal cord decreased [66].

### 4.3. Mechanisms Underlying the Loss of Central Morphine Analgesia in the ICGP Model

In the ICGP or IPGP models, morphine given via i.p. or i.c.v. routes did not show any analgesic actions, but this compound given through an i.t. or intra-plantar (i.pl.) route showed significant analgesic actions [56,66]. These findings suggest that the lack of morphine analgesia in these models is attributed to the plasticity in brain neuronal circuits but not to the opioid actions on sensory or spinal neurons. The lack of central morphine analgesia mechanisms is reminiscent of a previous study [88], in which morphine analgesic tolerance is mediated by the counter-balance mechanism through the upregulation of anti-opioid NMDA receptor function [89]. In more detail, the study described that the analgesic tolerance to repeated administrations of high doses of morphine was abolished in NR2A (a subunit of NMDA receptor)-KO mice and reversed by the genetic rescue of NR2A into the PAG [89]. Indeed, the lack of morphine analgesia in the ICGP model was recovered in NR2A-KO mice by the i.c.v injection of NR2A-specific antagonist or by the micro-injection of siRNA for NR2A into the PAG [89]. The authors speculate that the lack of morphine analgesia in the ICGP model is mediated by the activation of an anti-opioid system, which is caused by the excess release of endogenous opioids. This speculation may be supported by the report that the analgesia induced by mild swim stress was lost in β-endorphin-KO mice [90]. In other words, the lack of morphine analgesia in the ICGP model seems to be caused by endogenous opioid tolerance.

### 4.4. Potential Radical Treatments for FM

LPAR_1_ signaling is known to play key roles in the development and maintenance of NeuP and FM [63,64,65,91]. The repeated administrations of LPAR_1_ antagonist reversed established chronic pain, NeuP, and FM [56,62,92,93,94,95], suggesting that LPAR_1_ antagonist could be used as a radical treatment for chronic pain, unlike in the case with PGB or duloxetine, which could be used for symptomatic treatment. Because no clinically useful LPAR_1_ antagonist is available, we should consider other options to suppress chronic pain memory using available medicines. For this purpose, we propose an adrenergic α2 receptor antagonist-type anti-depressant, mirtazapine [55]. Repeated treatments with mirtazapine at 1 mg/kg for 1 week reversed the established hyperalgesia in the ICGP model, and the normal pain threshold lasted for at least 4 more days after stopping drug treatment. Unlike duloxetine or milnacipran, ICGP or IPGP was blocked by mirtazapine through i.c.v. but not i.t. [55,56]. The following study revealed that α2a receptor gene expression significantly increased only in the habenula among 16 brain regions at P5 after ICS. The micro-injection of siRNA for α2a receptor into the habenula significantly reversed hyperalgesia [55]. Although the underlying mechanisms of mirtazapine-induced anti-hyperalgesia remain elusive, it appears that they are different from the ones for duloxetine or milnacipran-induced beneficial actions. In addition, the repeated treatment with mirtazapine could be an example for radical treatment of FM. Another interesting example for radical treatment was found in the treatment with donepezil [79]. From the fact that FM patients show dry eyes and a dry mouth as symptoms, we first attempted the use of pilocarpine [79]. As ICGP was effectively blocked by pilocarpine through i.p. and i.c.v. but not i.t., we then decided to use an acetyl choline esterase inhibitor and donepezil, a drug used to treat Alzheimer’s disease, which easily penetrates the brain. When donepezil was used once daily from P5 to P10 after ICS at an effective and very low dose of 10 μg/kg (i.p.), the pain threshold was reversed to the control level until at least P18 without attenuation [79], suggesting that the pain memory was erased by donepezil. Thus, repeated treatment with donepezil could be used as another radical treatment for FM. PGB treatment could also be used as a candidate for the radical treatment of FM. In this trial, the P-glycoprotein inhibitor valspodar, which is supposed to inhibit drug excretion through the blood-brain-barrier, was co-administered with PGB at 1 mg/kg every other day from P5 to P11. The treatments reversed the pain threshold to the normal level until at least P19 without attenuation [73]. All these results indicate that the complete reversal of hyperalgesia may lead to erasing pain memory in the ICGP model.

### 4.5. Differential Mechanisms between Generalized Pain and Loss of Central Morphine Analgesia

The question of whether the generalized pain in FM models is directly associated with the lack of morphine analgesia is raised. Experimental evidence revealed that upon thermal, mechanical, and electrical stimuli to the paw and pressure stimuli to the deep muscle in the ICGP model, hyperalgesia was not affected in the μ-opioid receptor-KO mice [57]. This finding suggests that endogenous opioid functions are not associated with hyperalgesia in the FM model. This view was supported by the finding in a study using LPAR_1_-KO mice, which completely reversed hyperalgesia in the ICGP model without showing any central morphine analgesia [57]. Similarly, another supportive observation was found in the study of mirtazapine-induced anti-hyperalgesia and its underlying mechanisms [55]. In this study, the lack of morphine analgesia also remained after the reversal of hyperalgesia by the micro-injection of siRNA for α2a receptor into the habenula in the ICGP model, although the lack of morphine analgesia was reversed by repeated and systemic treatments with mirtazapine. This discrepancy may be explained by the finding that morphine analgesia was reversed by the i.c.v. administration of siRNA for the α2a receptor, suggesting that the α2a receptor in brain regions other than the habenula may participate in the lack of morphine analgesia or analgesic opioid tolerance.

### 4.6. Relationship to Immunity

Both the ICGP and IPGP animal models appear useful for further investigatory studies seeking to enhance FM diagnosis and potential pharmacotherapy. Regarding the potential use of these models for better diagnoses, further studies in relation to immunity are required in the future. Indeed, the relationship between FM and immunity has been frequently reported. Wallace et al. reported that interleukin (IL)-1R antibody and IL-8 levels in FM patients were significantly higher in sera when compared with healthy controls [96]. Similar observations have been shown in many reports [97,98,99,100]. Interestingly, there is a report that various chemokines and ILs, including IL-8, may be potential salivary biomarkers for an FM diagnosis [101]. Recently, there was an interesting study in which the plasma transfer of IgG from FM patients to mice showed several symptoms observed in FM patients [102]. A similar finding was also reported in the case of pain production in mice by IgM from patients with complex regional pain syndrome [103]. As IgG or IgM are secreted by B cells, which interact with other immune cells, such as macrophage or monocytes, it is interesting to speculate that peripheral LPAR_1_ signaling in macrophage/monocytes may make some contribution to the B cell activation. On the other hand, the levels of natural killer (NK) cells, which kill unwanted cells, are known to be negatively correlated with chronic pain conditions [104,105,106]. Of interest are the findings that NK cell numbers or cytotoxicity are increased by acute stress hormones or acute pain [107,108,109,110], whereas NK cell cytotoxicity is inhibited by chronic pain or chronic stress hormones through the hypothalamic-pituitary-adrenal axis [106,111,112]. There are reports that NK cells express LPAR_1_-3 [113,114], LPA decreases the cytotoxicity of NK cells [115], and secretes interferon-γ from NK cells [113]. However, it remains elusive whether LPAR signaling modulates chronic pain status through NK cell activities. Thus, studies to examine how these immunity-related activities correlate with pain status in ICGP/IPGP models would be important for future research.

### 4.7. Relationship to Small Fiber Neuropathy

The development of ICGP- and IPGP-type FM models is based on the view that central repeated (intermittent) stress is one of the major etiological events that causes FM. Indeed, pharmacotherapeutic studies show that hyperalgesia in these models was abolished by i.c.v. injection with small doses of pregabalin, mirtazapine, donepezil, and LPAR_1_ antagonist AM966 or by duloxetine via i.t. injection (Table 1). Even in the case of the AcGP model, with repeated injections of acidic saline into the gastrocnemius muscle, it was revealed that i.c.v. injections of microglia inhibitor minocycline abolished chronic hyperalgesia [116]. The brain mechanisms in AcGP have also been reported by using a different approach, where the micro-injection of mitogen-activated kinase inhibitor U0126 or phorbol 12,13-dibutyrate into the anterior nucleus of the paraventricular thalamus completely reversed hyperalgesia in the AcGP model [117]. Apart from such central mechanisms, a number of studies have reported that small fiber neuropathy is one type of peripheral etiological mechanism underlying FM [36,118,119]. Although there is a debate over the etiological meaning of small fiber neuropathy [35,120], it is interesting to examine whether ICGP and IPGP models display evidence for small fiber neuropathy. As small fiber neuropathy is recognized as a disorder of Aδ and C fibers, which are also characterized by several types of neuropathic pain [121,122,123], small fiber neuropathy should be considered as one of symptoms, and this is frequently observed in FM patients. In the Neurometer^®^ study using the ICGP model, however, hyperalgesia from Aδ-fiber stimulation was present, but with no significant change in C fiber stimulation (Table 1), indicating that a disorder of Aδ and C fibers is unlikely.

### 4.8. Limitations and Future Study

As mentioned above, hyperalgesia in ICGP and IPGP models continues for at least a couple of weeks after the last session of intermittent stress. However, most pharmacotherapeutic studies are performed 1–2 weeks after stress. When considering that, by definition, chronic pain, including FM, requires continued abnormal pain for ≥3 months, the beneficial actions of mirtazapine or donepezil (which are not approved for clinical use on FM) should be employed during the later stages in FM-like animal models to strengthen the validation for their clinical usage. As mentioned above, the study lacks detailed cellular and molecular mechanisms in the brain. Therefore, studies to examine the involvement of peripheral immunity, as well as immune-histochemistry and gene expression in pain-related brain loci (pain matrix), would represent important future topics of investigation. In this review, the authors intended to introduce the ICGP and IPGP models in terms of their patho-physiological and pharmacotherapeutic features in comparison to clinical ones. As we are using other FM-like models for our studies that are in progress, the comparison among different animal models should wait for the forthcoming studies.

### 4.9. Conclusions

The ICGP and IPGP models use intermittent cold and empathic psychological stress, respectively. Both models share similar patho-physiological (long-lasting and widespread pain; female-predominant pain) and pharmacotherapeutic features (negative effects with NSAIDs and morphine; positive effects with pregabalin and SNRIs) to each other, and the features mimic those in FM patients. Both models deal with the nature of pain memory because hyperalgesia persisted for several weeks, even after stopping the original stress. Therefore, this memory is likely maintained or reinforced by some feed-forward mechanisms, such as pain-induced pain through neural circuits [64] or peripheral immune activation [116]. This view is partly supported by the findings that LPAR_1_-KO mice or repeated treatments with an LPAR_1_ antagonist abolished hyperalgesia in the ICGP and IPGP models. A similar case of blocking pain memory was also observed with repeated treatments with mirtazapine and donepezil. These findings using the ICGP and IPGP models may pave the way for the development of medicines useful for the radical treatment of FM, although no supportive clinical evidence has yet been found. We cordially expect feedback from researchers using these models for the purpose of developing radical treatments for FM.

## Figures and Tables

**Figure 1 biomedicines-12-00056-f001:**
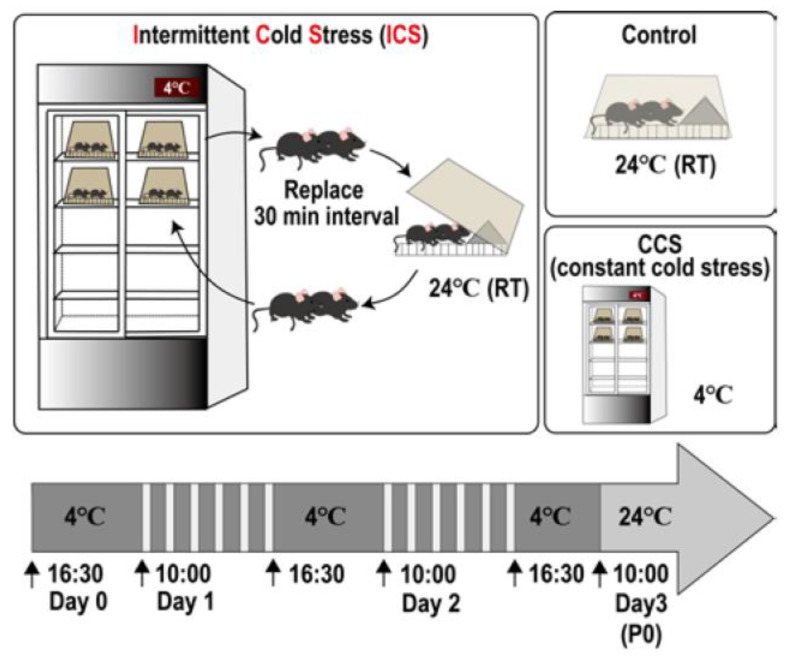
Schematic protocol of intermittent cold stress-induced generalized pain (ICGP) model.

**Figure 2 biomedicines-12-00056-f002:**
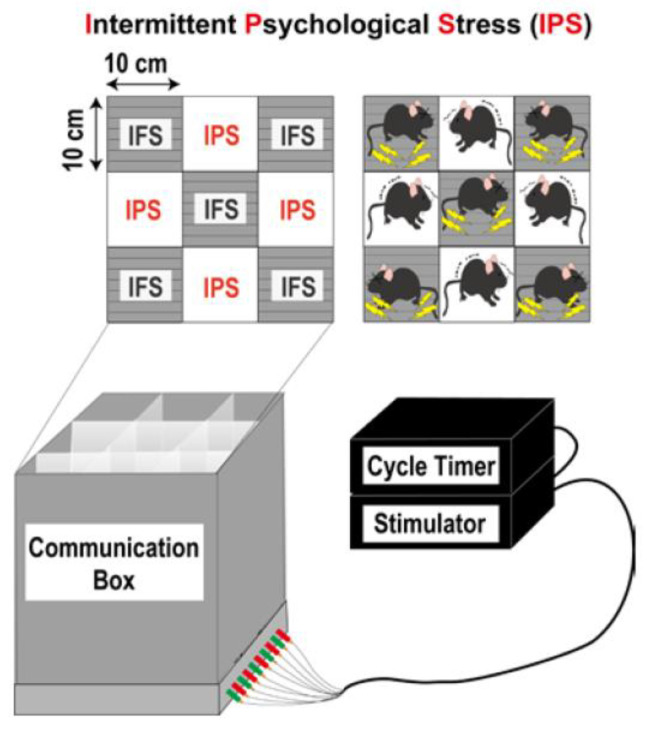
Schematic protocol of intermittent psychological stress-induced generalized pain (IPGP) model.

**Table 1 biomedicines-12-00056-t001:** Summary of pathophysiological and pharmacotherapeutic features in ICGP and IPGP model. Experimental evidence was found in the references, (**a**) Nishiyori and Ueda [51], (**b**) Nishiyori et al. [52], (**c**) Ueda and Neyama [56], (**d**) Nishiyori et al. [66], (**e**) Neyama et al. [57], (**f**) Mukae et al. [73], (**g**) Neyama et al. [55], (**h**) Mukae et al. [79].

Pathophysiology	ICGP	IPGP
Mechanical hyperalgesia	++	PWT (g)@P1–12 (**a**)	++	PWT@P1, P8 (**c**)
Thermal hyperalgesia	++	PWL (sec)@P15 (**a**)	++	PWL@P0–19 (**c**)
Electrical hyperalgesia	++−	250, 2000 Hz@P13, 12 (**h**)5 Hz@P13 (**h**)		
Depression-like behaviors	−	Tail suspension (**b**)	−	Tail suspensionForced swimming (**c**)
LPAR_1_-KO	Lost	PWL@P6 (**c**)	Lost	PWL@P1 (**c**)
Sex difference(after gonadectomy)		Female >> Male (**a**)		Female >> Male (**c**)
**Pharmacotherapy**	**ICGP**	**IPGP**
Diclofenac analgesia (i.p.)			−	PWL > 10 mg/kg (**c**)
Morphine analgesia (s.c.)	−	PWL > 3 mg/kg (**d**)	−	PWL > 1 mg/kg (**c**)
Morphine analgesia (i.c.v.)	−−	PWL > 0.3 nmol (**d,e**)250, 2000 Hz > 0.3 nmol (**e**)	−	PWL > 0.3 nmol (**c**)
Morphine analgesia (i.t.)	++++	PWL@P5 1 nmol (**d**)5 Hz@P5 1 nmol (**e**)		
Pregabalin (i.p.)	++	PWL 1 mg/kg (**f**)	++	PWL 1, 3 mg/kg (**c**)
Pregabalin (i.c.v.)	+++	PWL 1 μg (**f**)	+++	PWL 0.3 μg (**c**)
Pregabalin (i.t.)	+	PWL 10 μg (**f**)	+	PWL 3 μg (**c**)
Duloxetine (i.p.)			++	PWL 30 mg/kg (**c**)
Duloxetine (i.c.v.)			−	PWL > 3 μg (**c**)
Duloxetine (i.t.)			++	PWL 1 μg (**c**)
Milnacipran (i.t.)	++	PWL 0.1 μg (**b**)		
Mirtazapine (i.p.)	++	PWL 1 mg/kg (**g**)	++	PWL, PWT 1 mg/kg (**c**)
Mirtazapine (i.c.v.)	+++	PWL 1 μg (**g**)	+++	PWL 1 μg (**c**)
Mirtazapine (i.t.)	−	PWL > 1 μg (**g**)	−	PWL > 1 μg (**c**)
Donepezil (i.p.)	+++	PWL 10, 100 μg/kg (**h**)		
Donepezil (i.c.v.)	+++	PWL 10 μg (**h**)		
Donepezil (i.t.)	−	PWL > 10 μg (**h**)

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
