# Peer review of "Fibromyalgia Animal Models Using Intermittent Cold and Psychological Stress"

_biomedicines, 2023, doi:10.3390/biomedicines12010056_

Round 1

Reviewer 1 Report

Comments and Suggestions for Authors

The authors have written a review about fibromyalgia animal models using intermittent cold- and psychological-stress. Further, the authors have studied number of clinically used pain killing medicines and donezepil a drug used for symptomatic treatment of Alzheimer´s Disease and pharmacological tool compounds in their models and found that LPAR1 receptor plays a key role. Here the reader starts to wonder how translatable these models really are? Do they recapitulate human pathology associated with fibromyalgia?

Several years ago, it was reported that fibromyalgia patients suffer from small fiber neuropathy (PMID: 23748113), a finding that is replicated by several groups and was recently extended to autonomic nerves. Is there any evidence that intermittent cold- and psychological-stress can induce small fiber/autonomic neuropathy? Please discuss.

It is surprising that the authors did not discuss at all a recent study which shows that plasma transfer of IgG from fibromyalgia patients to mice recapitulates many of the fibromyalgia symptoms (PMID: 34196305). IgG is normally produced and secreted by B cells, which are interacting with other immune cells e.g. macrophages/monocytes, a cell type that is known to express LPAR1 receptor. Please, discuss this briefly.

A recent study reveals that NK cells are chronically activated and redistributed to the peripheral nerves in fibromyalgia patients (PMID: 34913882). Please discuss about possible role of LPAR1 in activation of NK cells in the pathogenesis of fibromyalgia. Are NK cells found in close proximity of the peripheral nerves in intermittent cold- and psychological-stress animal models?

Is there human genetic evidence that links LPAR1 to fibromyalgia?

Comments on the Quality of English Language

Minor editing is needed

Author Response

Authors’ replies to the Reviewer Report (Reviewer 1)

Thank you very much for these very important comments.

The authors have written a review about fibromyalgia animal models using intermittent cold- and psychological-stress. Further, the authors have studied number of clinically used pain killing medicines and donezepil a drug used for symptomatic treatment of Alzheimer´s Disease and pharmacological tool compounds in their models and found that LPAR1 receptor plays a key role. Here the reader starts to wonder how translatable these models really are? Do they recapitulate human pathology associated with fibromyalgia? 

According to these comments, we tried to improve the manuscript by adding some sentences to the 3-8 Conclusion, as seen in the highlighted parts of the revised manuscript (lines 682-684, lines 696-700).

Several years ago, it was reported that fibromyalgia patients suffer from small fiber neuropathy (PMID: 23748113), a finding that is replicated by several groups and was recently extended to autonomic nerves. Is there any evidence that intermittent cold- and psychological-stress can induce small fiber/autonomic neuropathy? Please discuss.

Regarding small fiber neuropathy, we added a highlighted new paragraph to the revised manuscript (3.7. Relationship to small fiber neuropathy, lines 648-668).

It is surprising that the authors did not discuss at all a recent study which shows that plasma transfer of IgG from fibromyalgia patients to mice recapitulates many of the fibromyalgia symptoms (PMID: 34196305). IgG is normally produced and secreted by B cells, which are interacting with other immune cells e.g. macrophages/monocytes, a cell type that is known to express LPAR1 receptor. Please, discuss this briefly.A recent study reveals that NK cells are chronically activated and redistributed to the peripheral nerves in fibromyalgia patients (PMID: 34913882). Please discuss about possible role of LPAR1 in activation of NK cells in the pathogenesis of fibromyalgia. Are NK cells found in close proximity of the peripheral nerves in intermittent cold- and psychological-stress animal models?

Regarding immunity issues (IgG, B cell, NK cells) and some LPAR1 relationships, we added a highlighted new paragraph to the revised manuscript (3.6. Relationship to immunity, lines 621-646).

Is there human genetic evidence that links LPAR1 to fibromyalgia?

LPAR1 mechanisms in the immunity system are described in lines 641-644. However, there is no human genetic evidence that links LPAR1 to fibromyalgia.

Reviewer 2 Report

Comments and Suggestions for Authors

This paper is a review of two animal models of fibromyalgia (FM), a chronic pain condition with various symptoms. The paper discusses intermittent cold stress (ICS) model: This model uses repeated exposure to cold temperature to induce long-lasting and generalized pain in mice. The model shows sex difference, lack of morphine analgesia, and involvement of lysophosphatidic acid receptor 1 (LPAR1) signaling. The model also responds to antidepressants, pregabalin, and donepezil, which are FDA-approved drugs for FM. The other is intermittent psychological stress (IPS) model: This model uses repeated exposure to psychological or empathic stress to induce long-lasting and generalized pain in mice. The model shows sex difference, corticosterone elevation, and involvement of LPAR1 signaling. The model also responds to antidepressants, pregabalin, and donepezil, as well as pilocarpine, a cholinergic agonist. The authors compared the two models in terms of their pathophysiological and pharmacotherapeutic features, and suggests that they could be useful for developing new medicines for FM. The paper also discusses the mechanisms of lack of morphine analgesia and potential radical treatments for FM. Specific comments:

1.          The abstract provides a good overview of the main findings and implications of the paper, but it could be improved by adding some information about the methods and results. For example, you could briefly describe how the ICGP and IPGP models were developed and validated, and what were the main pharmacological interventions and molecular mechanisms tested. You could also mention the limitations and future directions of the study.

2.          The introduction gives a comprehensive background on the definition, diagnosis, pathophysiology, and treatment of FM, as well as the existing animal models. However, it could be more focused and concise by avoiding unnecessary details and repetitions. For example, you could merge the paragraphs about the ACR and IASP criteria for FM, and summarize the main features of the nociplastic pain concept. You could also highlight the gaps and challenges in the current knowledge and practice of FM, and state the main objectives and hypotheses of your study more clearly.

3.          Some of the figures are not labeled properly or are too small to read. I recommend revising these figures to make them more clear and readable.

4.          The discussion section provides a good interpretation and integration of the main findings, but it could be more critical and balanced by acknowledging the limitations and uncertainties of the study. For example, you could discuss the validity and generalizability of the animal models, the potential confounding factors and sources of bias, the alternative explanations and hypotheses, and the ethical and practical implications of the study. You could also compare and contrast your findings with the existing literature, and identify the gaps and directions for future research.

5.          The conclusion section summarizes the main findings and contributions of the study, but it could be more concise and impactful by avoiding repeating the details and emphasizing the novelty and significance of the study. You could also provide some recommendations and suggestions for the clinical management and research of FM, based on your findings.

6.          The paper is an interesting and valuable contribution to the field of FM research, as it introduces and compares two novel and validated animal models of FM, and explores their pathophysiological and pharmacotherapeutic features and underlying molecular mechanisms. The paper also has some limitations and areas for improvement, as discussed above. Therefore, I recommend the paper for publication after minor revisions.

Author Response

Thank you very much for very important and constructive comments.

This paper is a review of two animal models of fibromyalgia (FM), a chronic pain condition with various symptoms. The paper discusses intermittent cold stress (ICS) model: This model uses repeated exposure to cold temperature to induce long-lasting and generalized pain in mice. The model shows sex difference, lack of morphine analgesia, and involvement of lysophosphatidic acid receptor 1 (LPAR1) signaling. The model also responds to antidepressants, pregabalin, and donepezil, which are FDA-approved drugs for FM. The other is intermittent psychological stress (IPS) model: This model uses repeated exposure to psychological or empathic stress to induce long-lasting and generalized pain in mice. The model shows sex difference, corticosterone elevation, and involvement of LPAR1 signaling. The model also responds to antidepressants, pregabalin, and donepezil, as well as pilocarpine, a cholinergic agonist. The authors compared the two models in terms of their pathophysiological and pharmacotherapeutic features, and suggests that they could be useful for developing new medicines for FM. The paper also discusses the mechanisms of lack of morphine analgesia and potential radical treatments for FM. Specific comments:

  1. The abstract provides a good overview of the main findings and implications of the paper, but it could be improved by adding some information about the methods and results. For example, you could briefly describe how the ICGP and IPGP models were developed and validated, and what were the main pharmacological interventions and molecular mechanisms tested. You could also mention the limitations and future directions of the study.

All these comments were answered in the revised Abstract (lines 12-35)

  1. The introduction gives a comprehensive background on the definition, diagnosis, pathophysiology, and treatment of FM, as well as the existing animal models. However, it could be more focused and concise by avoiding unnecessary details and repetitions. For example, you could merge the paragraphs about the ACR and IASP criteria for FM, and summarize the main features of the nociplastic pain concept. You could also highlight the gaps and challenges in the current knowledge and practice of FM, and state the main objectives and hypotheses of your study more clearly.

According to the reviewer’s comments, we merged the paragraphs about the ACR and IASP criteria (lines 59-63). The description to highlight the gaps and challenges in the current knowledge and practice of FM is shown in lines 74-77. The main objectives and hypothesis of our study are stated in lines 82-84, and 93-96.  

  1. Some of the figures are not labeled properly or are too small to read. I recommend revising these figures to make them more clear and readable.

Figures were amended.

  1. The discussion section provides a good interpretation and integration of the main findings, but it could be more critical and balanced by acknowledging the limitations and uncertainties of the study. For example, you could discuss the validity and generalizability of the animal models, the potential confounding factors and sources of bias, the alternative explanations and hypotheses, and the ethical and practical implications of the study. You could also compare and contrast your findings with the existing literature, and identify the gaps and directions for future research.

According to the reviewer’s comments, the statements of the validity and generalizability were described in lines 675-689. I am afraid the authors cannot answer the reviewer’s comments about the issues, potential confounding factors, sources of bias, alternative explanations, and hypothesis, since they are not fully specified. However, we will discuss these points after our ongoing experiments (lines 688-689). Regarding the ethical issue about animal experiments, we added the necessary description (lines 127-129, and lines 363-367). Regarding the practical implications of the study, we have stated the advantages in these models, which have similar pathophysiological (lines 498-500, 509-515, 528-530) and pharmacotherapeutic features as those reported with FM patients (lines 536-542, 545-549, 550-552). Regarding the intriguing findings that repeated treatments with mirtazapine or donepezil erased the pain memory, the authors stated their limitations (lines 584-587, 599-603). As the authors plan to perform other FM-like models, the comparisons in terms of pathophysiological and pharmacotherapeutic features should wait for the forthcoming studies (lines 688-689).

  1. The conclusion section summarizes the main findings and contributions of the study, but it could be more concise and impactful by avoiding repeating the details and emphasizing the novelty and significance of the study. You could also provide some recommendations and suggestions for the clinical management and research of FM, based on your findings.

According to the reviewer’s comments, the authors made the conclusion concise by summarizing main findings (lines 690-706).  

  1. The paper is an interesting and valuable contribution to the field of FM research, as it introduces and compares two novel and validated animal models of FM, and explores their pathophysiological and pharmacotherapeutic features and underlying molecular mechanisms. The paper also has some limitations and areas for improvement, as discussed above. Therefore, I recommend the paper for publication after minor revisions.

Round 2

Reviewer 1 Report

Comments and Suggestions for Authors

The authors have carefully addressed all concerns. Manuscript is now suitable for publication. Thank you!